# In Vitro and In Vivo Studies of Rumen-Protected Microencapsulated Supplement Comprising Linseed Oil, Vitamin E, Rosemary Extract, and Hydrogenated Palm Oil on Rumen Fermentation, Physiological Profile, Milk Yield, and Milk Composition in Dairy Cows

**DOI:** 10.3390/ani10091631

**Published:** 2020-09-11

**Authors:** Tae-Bin Kim, Jae-Sung Lee, Seung-Yeol Cho, Hong-Gu Lee

**Affiliations:** 1Department of Animal Science and Technology, Konkuk University, Seoul 05029, Korea; kim972678@daum.net (T.-B.K.); jslee78@konkuk.ac.kr (J.-S.L.); 2Institute of Research, Eugene Bio Ltd., Woncheon-dong, Suwon 16675, Korea; eugenebio@eugenebio.co.kr; 3Team of An Educational Program for Specialists in Global Animal Science, Brain Korea 21 Plus Project, Konkuk University, Seoul 05029, Korea

**Keywords:** omega-3 fatty acid, linseed oil, rumen bypass, milk, dairy cow

## Abstract

**Simple Summary:**

In this study, in vitro and in vivo analyses on the supplementation of rumen-protected microencapsulated fatty acids from linseed oil (MO) on rumen fermentation, physiological profile, milk yield, and milk composition in Holstein dairy cows were performed. We revealed that the supplementation of 2% MO incorporated into the diet is beneficial due to promoting omega-3 fatty acids in ruminant milk without negative/regressive effects on ruminal fermentation or the animal productivity of dairy cows.

**Abstract:**

The aim of the present study was to evaluate the effects of adding dietary rumen-protected microencapsulated supplements into the ruminal fluid on the milk fat compositions of dairy cows. These supplements comprised linseed oil, vitamin E, rosemary extract, and hydrogenated palm oil (MO; Microtinic^®^ Omega, Vetagro S.p.A, Reggio Emilia, Italy). For in vitro ruminal fermentation, Holstein–Friesian dairy cows each equipped with a rumen cannula were used to collect ruminal fluid. Different amounts (0%, 1%, 2%, 3%, 4%, and 5%) of MO were added to the diets to collect ruminal fluids. For the in vivo study, 36 Holstein–Friesian dairy cows grouped by milk yield (32.1 ± 6.05 kg/d/head), days in milk (124 ± 84 d), and parity (2 ± 1.35) were randomly and evenly assigned to 0.7% linseed oil (LO; as dry matter (DM) basis) and 2% MO (as DM basis) groups. These two groups were fed only a basal diet (total mixed ration (TMR), silage, and concentrate for 4 weeks) (period 1). They were then fed with the basal diet supplemented with oil (0.7 LO and 2% MO of DM) for 4 weeks (period 2). In the in vitro experiment, the total gas production was found to be numerically decreased in the group supplemented with 3% MO at 48 h post in vitro fermentation. A reduction of total gas production (at 48 h) and increase in ammonia concentration (24 h) were also observed in the group supplemented with 4% to 5% MO (*p* < 0.05). There were no differences in the in vitro fermentation results, including pH, volatile fatty acids, or CH_4_ among groups supplemented with 0%, 1%, and 2% MO. The results of the in vitro study suggest that 2% MO is an optimal dosage of MO supplementation in cows’ diets. In the in vivo experiment, the MO supplement more significantly (*p* < 0.01) increased the yield of total w3 fatty acids than LO (9.24 vs. 17.77 mg/100 g milk). As a result, the ratio of total omega-6 to omega-3 fatty acids was decreased (*p* < 0.001) in the MO group compared to that in the LO group (6.99 vs. 3.48). However, the milk yield and other milk compositions, except for milk urea nitrogen, were similar between the two groups (*p* > 0.05). Collectively, these results suggest that the dietary supplementation of 2% MO is beneficial for increasing omega-3 fatty acids without any negative effects on the milk yield of dairy cows.

## 1. Introduction

The fat content in ruminant feed is generally about 2.5% to 3.5% of dry matter (DM). Most oilseeds fed to dairy cows are soy, corn, and cotton, followed by canola, rapeseed, flaxseed, safflower, and sunflower. Some plant and animal by-products are also used in feed as sources of fat and oil. In the dairy cow diet, fat supplements are primarily used as an energy source to meet the cows’ high energy and nutritional needs. However, increasing the fat content of feed has several negative effects on ruminal microbial growth, digestibility, and other characteristics of ruminants [1]. These negative effects of fat supplementation on ruminal characteristics are more severe when adding unsaturated fatty acids, such as omega-3 and omega-6 fatty acids, than when adding saturated fatty acids. To prevent these problems of adding fat and oil to the diet, a type of ruminal protective fat has emerged. This type of fat can partially pass through the rumen, reach the small intestine, and become absorbed. Several techniques, such as the use of calcium salts of fatty acids, have already been developed to commercialize ruminal protective fat [2].

It is most common to increase the content of omega-3 fatty acids in feeds by increasing the omega-3 fat in rumen, especially in cows’ milk. Linseed oil is often used as a source of fat for the promotion of α-linolenic acid, an omega-3 fatty acid. Unsaturated fatty acids, such as omega-3 fatty acids, can enter the rumen and undergo extensive biohydrogenation, thus affecting a wide range of microorganisms and resulting in the loss of omega-3 fatty acids. Linseed oil is mostly composed of unsaturated fatty acids. It is known to inhibit microbial metabolism due to its negative influence on digestibility. Although this oil is a major source of omega-3 fatty acid, it also has adverse effects, such as decreasing dairy cow dry matter intake (DMI) [3]. On the other hand, hydrogenated palm oil is mostly composed of saturated fatty acids. It is known that supplementing a diet with hydrogenated palm oil does not have adverse effects on DMI but does increase the energy content in feed, thus improving milk yield and milk fat content [4]. Therefore, we hypothesized that feeding cows rumen-protected microencapsulated essential fatty acids using linseed oil coated with hydrogenated palm oil could increase the omega-3 fatty acids in milk without adversely affecting the ruminal fermentation, DMI, or milk characteristics of dairy cows. To test our hypothesis, rumen bypass microencapsulated fatty acids (MO; Microtinic^®^ Omega, Vetagro S.p.A, Reggio Emilia, Italy) from linseed oil were introduced to an in vitro rumen model and an in vivo trial. Thus, the objective of this study was to determine the optimal dosage of MO by using an in vitro rumen fermentation assay. Based on the findings of the in vitro study, we confirmed the effects of adding MO to the diets of lactating dairy cows on their ruminal fermentation, physiological traits, milk yield, and milk composition.

## 2. Materials and Methods

All experimental procedures involving animals were performed according to the Animal Experimental Guidelines provided by the Animal Care and Use Committee of Konkuk University, Republic of Korea (approval number: KU18182).

### 2.1. Experimental Materials and Procedures for the In Vitro Experiment

A Holstein dairy cow weighing 735 kg at 30 months of age equipped with a rumen cannula was used to conduct the sampling of ruminal fluid. The animal was fed alfalfa and concentrate (a ratio of 6:4) twice a day at 0900 and 1600 h. Water was supplied ad libitum. Ruminal fluid was collected by filtering with two layers of cheese cloth through the rumen cannula two hours before feeding. The collected ruminal fluid was kept in a 2-L thermos bottle filled with preheated CO_2_ gas, while oxygen (O_2_) was completely removed. This fluid was maintained at 39 °C and then moved to the research lab. After removing the feed particles with a vacuum pump, the supernatant was used for analysis.

The preparation of MO was obtained from Vetagro S.p.A. The material was prepared via a patented spray-cooling microencapsulation process. Briefly, after heating the hydrogenated palm oil, linseed oil was mixed with hydrogenated palm oil and an active principle. A mixture of materials was moved to a chilling chamber and then cut to a size of 1000 to 1500 microns. Finally, the MO was sealed and stored at 4 °C. The ingredients of MO were linseed oil (35%), vitamin E (0.5%), rosemary extract (0.3%), and hydrogenated palm oil (64.2%). The main components of MO were crude fat (95%) and ash (5%).

There were two separate in vitro fermentations. For the analysis of ruminal pH, gas, volatile fatty acids (VFA), and ammonia-N, the total mixed ration (TMR) was dried in a drying oven (JS-CO2-AT100, Johnsam Co. Ltd., Bucheon, Korea) at 60 °C for 24 h. Forage and concentrate were crushed with a 2-mm screen and used for the TMR at a ratio of 6:4. The TMR (0.2 g) and 20 mL of anaerobic McDougall buffer [5] were added to a 60 mL flask with 10 mL of ruminal fluid. Before adding 30 mL of culture solution to the culture flask, 0% to 5% MO (% Microtinic Omega of DM, arranged from Vetagro S.p.A, Italy) was added to the diet. Each ruminal sample was kept under maximum anaerobic conditions by flushing the CO_2_ gas and capping with a rubber cap to prevent the leakage of gas as much as possible. The experiment was conducted in three replicates of six treatments for different time intervals (0, 12, 24, 48 h) in a shaking incubator (SI-900R, Jeio Tech Co. Ltd., Daejeon, Korea; 100 rpm) at 39 °C. For each time point, we took samples from separate flasks. Because we used the same four flasks (same culture) for sampling at the time points of 0, 12, 24, and 48 h, the time of sampling should not have had any effects on the samples or related measures. All flasks had the same sizes and conditions. Hence, time was not needed in the model. Measurement of the total gas was conducted after uncapping the medium with the aluminum cap using a 50-mL gas cylinder (Habdong Co., Anyang, Korea) at each fermentation time [6]. To analyze digestibility and long chain fatty acids (LCFAs), the same TMR was grinded, and 1.5 g was placed into ANKOM filter bags (ANKOM Tech., Macedon, NY, USA). Along with 450 mL culture solution (same ratio with above), three independent bags were placed in one bottle per treatment for each fermentation time with 0.7 linseed oil or 2% MO (as DM basis). The sampling time was the same as that of the first fermentation, and after each sample extraction, we washed the sample using water and always put the samples into a desiccator for protection from contamination and air.

### 2.2. Analysis

After each incubation, the total gas and methane were detected. A portion of the ruminal sample was then used for the analysis of pH, VFA, ammonia-N, digestibility, and LCFAs. Before measuring the total gas, 0.3 μL of the gas in each flask was carried by the GC cylinder to a GC injector (injection volume: 0.3 μL). Methane (%) was then detected using a GC/TCD (HP 6890, Hewlett Packard Co., Palo Alto, CA, USA) equipped with a HP-PLOT-A column (dimension: 30 m × 0.32 mm × 20.00 μm) according to Owen et al. [6]. A culture solution of 2 mL was collected for the ammonia (1 mL) and VFA (1 mL) analyses. All samples collected were kept frozen at −20 °C until they were analyzed. The ammonia concentration was determined via the method of Fawcett and Scott [7] using a spectrophotometer (Model 680, BIO-RAD, Hercules, CA, USA). The culture solution of 1 mL was mixed with 0.1 mL 25% phosphoric acid. Then, 0.2 mL pivalic acid solution (2%, *w*/*v*) was added as an internal standard. The mixed solution was centrifuged at 15,000× *g* for 15 min. The supernatant was used to determine the concentration and composition of the VFA using a gas chromatograph mass spectrometer (HP 6890, Agilent Technologies, Santa Clara, CA, USA).

The analysis of fiber digestibility as a neutral detergent fiber (NDF) and acid detergent fiber (ADF) was conducted using an ANKOM filter bag (ANKOM Tech.), and the NDF and ADF contents were estimated based on the following equations [8]:

NDF, ADF (%) = [dried residue − (filter bag weight × C)] / sample weight × 100
(1)
where ADF is the dried sample weight after NDF processing and C means correction (dry blank weight/original blank bag weigh):

NDF, ADF (g) = sample weight (g) × DM (%) × NDF or ADF (%) / 10,000
(2)

NDF, ADF digestibility (%) = NDF or ADF (g) of zero time / NDF or ADF (g) of each time
(3)

Briefly, each filter bag including 0.45 g feed was immersed in acetone for 10 min, dried for 10 min, and then put into an ANKOM fiber analyzer (ANKOM 200, ANKOM Tech.). After adding the NDF or ADF solution, 4 mL of α-amylase (not included in the ADF analysis), and 20 g of anhydrous sodium sulfate, the mixture was heated at 100 °C for 75 min. After heating, the sample was washed with 2 L of distilled water at 70 °C for 12 min. Each sample was then immersed in acetone for 10 min, dried at room temperature for 10 min, and further dried in a drying oven for 48 h at 60 °C before weighing. Crude protein (CP) was calculated by multiplying the CP percentage (% N) by 62.5, which was derived from an elemental analyzer (EA 1110, CE instruments Ltd., Hindley Green, Wigan WN2 4HF, UK).

CP digestibility (%) = 100 − [CP (g) of each time / CP (g) of zero time] × 100.
(4)

Briefly, 3 mg of feed and 0.1 mg of sulfanilamide were added to each tin solid capsule (PN 240 06400, Thermo Scientific Inc., Waltham, MA, USA) using sulfanilamide as a standard substance at an ultrafine scale. Ether extracts (EE; crude fat) were analyzed after ether extraction using an extraction system (SOXTHERM, C. Gerhardt GmbH & Co. KG, Königswinter, Germany). This analysis was calculated based on the weight difference of the sample before and after ether extraction.

EE digestibility (%) = 100 − [EE (g) of each time / EE (g) of zero time] × 100
(5)

The chemical analyses for the resulting samples were conducted in accordance with the standard methods for the association of official analytical chemists (AOAC) [9].

For LCFA analysis, 20 mL of the solution was also collected at each incubation time and freeze dried. Lipids were then extracted using Folch’s solution and analyzed using a gas chromatograph/flame ionization detector (FID) (HP 7890 series GC System, Agilent technologies Inc., Santa Clara, CA, USA) equipped with an Sp-2560 capillary column (dimension: 100 m × 0.25 mm × 0.2 μm film thickness). Cis-9, trans-11 conjugated linoleic acid (CLA), and trans-10, cis-12 CLA (Sigma–Aldrich Co, St. Louis, MO, USA) were used as an internal standard.

### 2.3. Animals and Design for In Vivo Experiment

First, the absorption of MO in blood was verified. The measurement of α-linolenic acid (ALA) flow in the blood was conducted using a catheter. To measure the ALA (omega-3 fatty acid) absorbed into the blood after feeding with 2% MO, a Surflo IV Catheter (Terumo Corporation, Shibuya-ku, Japan) was installed into the jugular veins of four cows. Blood sampling was then carried out every 2 or 4 h (*n* = 4) using a serum separator tube (Vacutainer, Becton Dickinson, Franklin Lakes, NJ, USA). Blood samples were then immediately separated into serum and stored at −20 °C until fatty acid analysis.

After verification of absorption, a total of 36 Holstein dairy cows (milk yield: 32.1 ± 6.05 kg/d/head, DIM: 124 ± 84 d, and parity: 2 ± 1.35 calving number) were randomly and evenly assigned to the LO (0.7% linseed oil as DM basis) and MO (2% MO as DM basis) groups and studied for 8 weeks. Each pen (6 cows/pen) in the feedlot was equipped with automatic water troughs. Using a completely randomized design, these two groups were fed only with the basal diet for 4 weeks (period 1). They were then fed the basal diet supplemented with oil (0.7% and 2% of DM) for 4 weeks (period 2). The dairy cows were fed the TMR (78%) (Seowon Feed Co., Ltd., Pyeongtaek-si, Korea), silage (6%), and concentrate (16%). The experimental diets were formulated to meet or exceed National Research Council (NRC) recommendations [10]. Water was provided ad libitum. Diets were offered in equal amounts once daily (0830 h). Feed consumption was recorded daily by weighing the feeds offered to and refused by the cows. The DMI was then determined based on the kg of feed consumption and the moisture of feed per cow in each group. The ingredients and chemical compositions of the experimental diet are presented in Table 1. We received the MO and linseed oil present in the MO from Vetagro S.p.A. The diet was provided once daily at 0800 on a DM basis. After 0.7% LO (linseed oil; Vetagro S.p.A, Italy) or 2% MO (Vetagro S.p.A) was mixed with the vehicle (corn gluten powder), the vehicle was added to the TMR by top dressing at period 2. The experiment was designed to determine the differences between periods (period 1 and period 2) and between groups (LO group and MO group).

### 2.4. Analysis of Milk Composition and Milk Fatty Acids

During all periods, milk was collected twice a day (0300 and 1500) individually. The total milk yield was then calculated, and its mean for each period was determined. After mixing the collected milk (100 mL) in the morning and afternoon, the milk samples were stored at 4 °C for milk composition analysis while leaving 30 mL milk (subsamples) for the analysis of milk fatty acids. Subsamples were stored at −20 °C for fatty acid analysis. The milk composition and milk fatty acids were analyzed on the first day at 0, 4, and 8 weeks.

Milk composition was measured with a MilkoScan (CombiFoss FT+500 S/H, Hillerød, Denmark). Milk lipids were extracted and detected using the same methods described in the in vitro experiment. The analysis conditions were Sp-2560 capillary column (dimension: 100 m × 0.25 mm × 0.2 μm, film thickness); injection, split 30:1; heater, 255 °C; pressure, 32.64; total flow, 39.5 mL/min; split flow, 36.0 mL/min; injection volume, 1.0 μL; carrier gas, helium 1.2 mL/min; oven program, 70 to 100 °C at 5 min (hold: 2 min), 100 to 175 °C at 10 min (hold 40 min), 175 to 225 °C at 5 min (hold 40 min); detector, FID System; and heater, 260 °C (H2 flow: 40 mL/min, air flow: 400 mL/min). The CLAs, omega-3 (C18:3n3, C20:3n3, C20:5n3, and C22:6n3) and omega-6 (C18:2n6t, C18:2n6c, C18:3n6, C20:3n6, and C20:4n6) fatty acids (Sigma–Aldrich Co) were used as an internal standard.

### 2.5. Analysis of the Complete Blood Cell Count (CBC) and Metabolite Profile Test in the Blood

Blood sampling for blood cell counting [11] and metabolite analysis [12,13] was conducted for each cow via the jugular vein at 3 h before feeding on the first day at 0, 4, and 8 weeks. In brief, blood was collected into an EDTA tube (BD 367844 Vacutainer, Becton Dickinson, NJ, USA) and then subjected to a CBC test using an HM2 (VetScan HM2 Hematology System, Abaxis, Union City, CA, USA). The plasma portion of the blood was prepared using a heparin tube (BD 367874 Vacutainer, Becton Dickinson, NJ, USA) and centrifuged at 3000 rpm for 15 min. The sample was then used to determine the blood metabolites using a chemical analyzer (Furuno CA-270, Nishinomiya, Japan).

### 2.6. Statistical Analysis

The results obtained from in vitro and in vivo experiments were subjected to a least squares analysis of variance according to the ANOVA procedure and the mixed procedure of IBM SPSS Statistics 21, respectively. In the in vitro experiment, linear and quadratic contrasts were used to determine the effect of increasing the amount of LO on the response variables. To rank the treatment means within a significant F test, a Duncan multiple range test was used. In the in vivo experiment, the initial data (not shown) of the milk composition and fatty acids were used as the covariates of the experimental data. In the results of the milk and blood components, the values of “Before” and “After” (meaning periods before feeding supplemental oil (the average of 0 and 4 w data) and after (the average of 4 and 8 w data)) were compared between the LO and MO simultaneously. Statistically significant differences were considered at *p* < 0.05. Differences among means with 0.05 < *p* < 0.10 were considered to show a significant difference. For a completely randomized design, the statistical model was:

Υ_ij_ = μ + Τ_i_ + ε
(6)
where μ is the overall average, Τ_i_ is the average for treatment *i*, and ε is a random error.

## 3. Results

### 3.1. In Vitro Fermentation

We investigated the changes in pH, total gas, and methane in ruminal fluid co-cultured with various concentrations of MO using an in vitro rumen model (Table 2).

Ruminal pH was not significantly affected (0.05 < *p* < 0.1) by the supplementation of 1% to 5% of MO in the ruminal fluid. It was observed that methane (%) proportions in the treatments of 3% to 5% MO addition were higher (*p* < 0.05) than those in the control at 12 h. Groups supplemented with 4 to 5% MO caused higher methane (%) production than the control at 24 h. The yield of methane (mL) was not significantly different between the control and treatment groups. However, the total gas production at 48 h showed a numerical decrease in the group supplemented with 3% MO. In groups supplemented with 4 to 5% MO, methane (mL) production was significantly decreased (*p* < 0.05). During rumen fermentation, although an increase in the concentration of ruminal ammonia-N via the supplementation of MO was observed initially, this phenomenon changed after 24 h of incubation. Notably, ammonia-N was numerically increased initially in the groups supplemented with 0% to 3% MO and then decreased rapidly in groups supplemented with 4% to 5% MO at 24 h. Collectively, these results suggest that 2% MO is an optimal dosage to be added to animal diets because it can increase the absorption of fats added to the small intestine.

In the ruminal fluid, the total VFA production in the supplementation group was not significantly different from that of the control group (Table 3). However, the ratio of acetate to propionate decreased in the groups supplemented with MO at 48 h after incubation (*p* < 0.05).

Based on the rumen, we confirmed that there was no significant (*p* > 0.1) difference in feed digestibility between treatment 1 (0.7% LO) or treatment 2 (2% MO) and the control (Table 4).

Regarding the variation in ruminal LCFA after each culture during rumen fermentation, higher amounts of ALA were observed in treatment group 1 (0.7% LO) and treatment group 2 (2% MO) than in the control (Table 5).

There were no statistically significant differences in the amount of ALA between treatment groups at 0 h of incubation time. After incubation, a higher amount (about 2-fold or more) of ALA in the ruminal fluid was observed in treatment group 2 than that in treatment group 1. In addition, stearic acid was present in a much higher (*p* < 0.05) amount in the control and LO groups than in the MO group at 12 and 24 h after incubation. The yield of CLA was significantly (*p* < 0.05) higher in the MO group than in the control or LO groups.

### 3.2. In Vivo Trial

The results of the average absorption rates of major fatty acids (linoleic acid, α-linolenic acid) revealed that the ALA content in blood was significantly (*p* < 0.05) increased after feeding with 2% MO (as DM basis) compared to that in the control from about 14 h (Figure 1).

There was no significant difference in the absorption rate of linoleic acid between the control and MO groups.

The effects of LO or MO supplementation to the basal diet on the DMI, milk compositions, physiological traits, and milk fatty acid compositions in daily cows were also evaluated. The results of the feed intake and milk composition showed that the DMI decreased (*p* < 0.001) via LO (Table 6).

With a decrease in DMI, an increase in milk protein (%) was observed (*p* = 0.034) in the LO group compared to the MO group after supplementation. However, the yield of milk protein (kg) was not significantly different between the LO and MO groups. The milk urea nitrogen (MUN) content was found to be decreased (*p* < 0.001) in the two groups after supplementation. The MUN value of the MO group was much lower than that of the LO group.

Blood samples collected at the beginning of the test (0 weeks), at the mid-test (4 weeks), and at the end of the study (8 weeks) were analyzed for their hematological and metabolic parameters. The results are shown in Table 7.

Although granulocyte, red blood cell (RBC) and hematocrit counts were significantly decreased (*p* = 0.004 and *p* = 0.002, respectively) after supplementation compared to those before supplementation, they were still within normal ranges. Conversely, the mean corpuscular hemoglobin (MCH) and mean corpuscular hemoglobin concentrations (MCHC) were outside of their normal ranges (*p* < 0.001). The blood metabolite analysis showed that blood urine nitrogen (BUN) was significantly (*p* = 0.002) decreased, while total cholesterol (TCHO) was increased (*p* = 0.052) in both the MO and LO groups compared to the control group. The non-esterified fatty acids (NEFAs) in blood were increased (*p* < 0.001) in the MO group compared to those in the LO group.

Regarding the compositions of milk fatty acids, omega-3 fatty acids (C18: 3n3, ALA) showed increases in both groups with MO supplementation (Table 8).

However, a much greater increase was observed (*p* < 0.001) in the MO group compared to that in the LO group (5.83 vs. 13.98 mg/100 g milk). Linoleic acid (C18: 2n6, LA), the most abundant omega-6 fatty acid, did not show a decrease in the LO group, although it was decreased (*p* = 0.005) in the MO group. CLA, another functional fatty acid, did not show statistically a significant difference in its amount between the MO and LO groups. However, a numerical increase was observed in the MO group compared to the LO group (10.00 vs. 11.72 mg/100 g milk). Therefore, the ratio of total omega-6 to omega-3 fatty acids showed a decrease (*p* < 0.001) in both groups. This ratio showed a greater decrease (*p* < 0.001) in the MO group compared to the LO group (6.99 vs. 3.48).

## 4. Discussion

The portion of linseed oil in MO (linseed oil coated with hydrogenated palm oil) used in this study was 35%. Its main fatty acids were palmitic acid and α-linolenic acid. It has been reported that the protection level of linseed oil from rumen microbes is 99% for 8 h. Thus, it is expected to have a higher coating effect than other rumen bypass omega-3 materials.

### 4.1. In Vitro Experiment

The effects of fat supplementation in the diet on ruminal pH vary widely. The type of fat source and fat composition can also affect ruminal pH in different ways. Ivan et al. [14] reported that feeding vegetable oils to ruminants can cause some changes in the ruminal pH because a large amount of unsaturated fatty acids is produced in the rumen, resulting in changes to ruminal fermentation patterns. In the present study, there was no significant difference in pH among all treatment groups, although a decrease in pH was observed alongside an increase in the amount of added MO (Table 2). Moreover, from 12 h, the numerical difference in ruminal pH between the highest and the lowest pH (5.80 vs. 5.75) was only 0.05 in the MO treatment groups, indicating that MO might not have a negative effect on microbial fermentation.

Typically, in vitro ruminal experiments have shown a negative correlation between total gas production and methane production in the case of fat supplements [15,16]. Fat can cause lipolysis and can simultaneously decrease methane production and increase total gas production by affecting ruminal microbes. In the present study, there was no significant difference in methane yields among treatment groups (Table 2). However, the total gas production was drastically decreased in the MO treatment group (decreased by 3% or more with MO addition at 48 h incubation time). Although MO is a rumen-protected fat, the decrease in total gas production after treatment with a supplement containing MO was similar to the results of Getachew et al. [15], which showed a reduction in gas production after treatment with corn oil. Considering that oil supplements can inhibit ruminal microbial metabolism, an overdose of bypass fat supplements may deteriorate microbial fermentation and result in decreased total gas production. However, a supplement of up to 2% MO did not decrease total gas production, suggesting that linseed oil was perfectly encapsulated by hydrogenated palm oil.

There was no significant difference in the amount of VFA between the control and MO treatments until 24 h of culture (Table 3). However, the ratio of acetate to propionate (A:P ratio) was found to be further reduced in the MO treatment group compared to that in the control group at 48 h after incubation. Generally, when the energy content or fat content of feed increases, the amount of propionate also increases, while the amounts of acetate and butyrate are simultaneously decreased due to the changing ruminal environment affected by the increased fermentation of organic matter [15]. The A:P ratio may have decreased because higher fat content stimulated greater ruminal fermentation of organic matter. Moreover, the fat eluted from MO might have affected this ratio since the bypass rate of MO was 99% until 8 h.

Alterations in the ammonia-N concentration were observed after the addition of MO at all incubation times (Table 2). Notably, at an addition level of more than 3%, the results showed a numerical increase in the ammonia-N concentration at 24 h. According to Machmüller et al. [16], the ruminal defaunation effect caused by fat sources can promote the growth of the bacteria involved in microbial protein synthesis and reduce the ammonia-N content in the rumen. Borchers [17] also reported that the addition of essential oil to in vitro ruminal fluid can result in the accumulation of amino acid N and a reduction in the ammonia N concentration, suggesting that oil supplementation can inhibit deamination. On the other hand, the results of ammonia-N in this study using MO were different from those of previous studies. Firkins [18] reported that microbial protein synthesis is the most active in neutral pH for fiber digestion. Considering all these results, the tendency of ruminal pH to decrease after MO addition in the present study (which is different from previous studies, although the fat source added was the same between the present study and previous studies) might have negatively affected the synthesis of microbial proteins in the rumen, resulting in an increase of ammonia-N in the rumen. However, ammonia-N was not affected by MO at a dosage up to 2%. This might be due to the coating effect of hydrogenated palm oil.

Therefore, it can be concluded that the addition of 2% MO can most strongly increase the absorption of fat in the intestine. We then determined the feed digestibility and LCFAs with the supplementation of 2% MO and 0.7% linseed oil (LO) that had the same level of ALA to determine whether the differences were derived from the coating. The analysis of feed digestibility and LCFAs, including the ALA content in ruminal fluid, confirmed that digestibility was not affected by either supplementation (Table 4). The ALA content in the MO supplementation group was significantly higher than that in the LO treatment group during the ruminal fermentation period (Table 5). Lower stearic acid levels were observed in the MO group than in the LO or control groups during incubation, suggesting greater protection of LA and ALA in MO. Moreover, the control and LO groups showed a more rapid appearance and disappearance of CLA (an intermediate product between LA and stearic acid) than the MO group, supporting the successful encapsulation of linseed oil in MO.

### 4.2. In Vivo Experiment

The absorption rate of omega fatty acids (ALA and LA) was determined to confirm their transfer from the feed to the blood. As mentioned earlier, 2% MO (as DM basis) was selected because MO mostly affected the absorption of fatty acids contents while remaining safe for ruminal characteristics. This was verified by the increase in ALA from 14 h after feeding with a diet supplemented with 2% MO. As expected, the DMI was significantly lower in the LO-supplemented group than that in the MO group (Table 6). Generally, the addition of unsaturated fatty acids has a toxic effect by inhibiting microbial metabolism in the rumen, decreasing NDF digestibility, and yielding a lower DMI in feeds containing fat [1]. Several studies have shown that linseed oil can reduce the palatability of feeds due to its adverse effect on digestibility, further reducing DMI [3]. In the present study, the DMI differences between the LO group and the MO group were most likely due to differences in the types of fatty acids introduced into the rumen. Including MO in the diet has a rumen bypassing effect for unsaturated fatty acids by exposing hydrogenated palm oil (the saturated fatty acid part before the internal oil is released) to ruminal microbes. On the other hand, LO is mostly a liquid oil made of unsaturated fatty acids. The liquid part (unsaturated fatty acid) of LO might have an adverse effect on the microbial environment in the rumen, which may negatively affect ruminal digestibility as described above, thereby decreasing DMI. The milk urea nitrogen content was found to be decreased in both oil groups in the present study. The nitrogen in milk is the result of the remaining nitrogen being absorbed into the blood and then transferred to the mammary gland after metabolism of the protein or nitrogen compounds in feed used for the synthesis of microbial proteins in the rumen [19]. Beauchemin et al. [20] reported that the addition of fat in ruminants could not only be used as an energy source for their protein synthesis in rumen microbes but can also affect ammonia-N in the rumen or the nitrogen content in the milk. In addition, the nitrogen content in milk is known to be greatly influenced by the lactating period of dairy cows. The MUN peak due to an energy balance imbalance in early lactating dairy cow declines with the passage of time [21]. In the present study, we assumed that the effect of the milking period was more influential than the effect of fat addition on MUN reduction (Table 6). Particularly, the MUN of the MO group was lower than that of the LO group. This might be because the exposure of unsaturated fatty acids in the rumen resulted in the inhibition of ruminal microbial activity in the LO group.

The complete blood cell count (CBC) is a physiological index of animals [22]. It was used to determine the safety of the experimental additives used in this study for animals. As mentioned before, the granulocyte, red blood cell, and hematocrit counts were significantly decreased (*p* < 0.01) in both the MO and LO groups (Table 7). However, their values after decreasing were still within normal ranges. Unexpectedly, the blood levels of MCH and MCHC were increased in both the MO and LO groups. Few studies have reported the correlation between hemoglobin parameters and fat supplementation. Red blood cells were in a normal range after fat supplement, and anemia was not observed in any experimental period. Therefore, these increases of MCH and MCHC were considered to have no significant effects on animal physiology. Similar to the results of the present study, several authors have reported that lipids in feed can affect the BUN and lipid levels in blood [23,24,25]. For the results of blood metabolites, BUN decreased for reasons similar to those of the decreased MUN shown in Table 6. Likewise, BUN decreased because of changes in nitrogen—not only due to fat utilization as an energy source to synthesize microbial proteins but also due to the flow in the lactating period. As expected, in both the MO and LO groups, TCHO was increased due to the addition of fat in the feed, especially in the MO group (51.2 mg/dL vs. 25.7 mg/dL, *p* = 0.052), mainly because the amount of fat added to the feed was higher in the MO group than that in the LO group (4.80 vs. 3.59 ether extract, %). Conversely, the NEFA content in the blood was decreased in the LO group, which might have been due to decreased DMI in the LO group. Collectively, it can be considered that feeding MO protected by saturated fatty acids can induce a stable intake of fat content in ruminants, thus having a greater influence on the lipids absorbed by the small intestine than feeding linseed oil.

The stearic acid in milk is an 18-chain saturated fatty acid without a double bond. It is derived from transference to the mammary glands after the biohydrogenation of 18-chain fatty acids with double bonds (e.g., LA or ALA) via microbes in the rumen or the induction of stearic acid in feed. As expected, the supplementation of stearic acid in LO and MO increased the stearic acid levels in milk (Table 8). Particularly, it yielded a greater increase in the MO group than in the LO group. This occurred because the amount of stearic acid in the MO diet was higher than that in the LO diet (Table 1). Furthermore, in the LO group, the addition of linseed oil decreased the DMI. Thus, the decrease of stearic acid content in milk was thought to be due to the lower intake of stearic acid. The total SFA was lower, while the total MUFA was higher in the MO group than in the LO group. Due to the ruminal protection effect of hydrogenated palm oil, the unsaturated fatty acids in linseed oil are rarely released or digested and almost entirely absorbed by the small intestine, resulting in a lower value of SFA/UFA in the MO group than in the LO group [26]. There was no statistically significant difference in CLA content in milk, although the CLA content numerically decreased in the LO group but increased in the MO group compared to the control. According to Doreau and Ferlay [27], the addition of linseed oil in the diets of ruminants can generally induce the isomerization of ALA and LA in the rumen via ruminal microbes, leading to CLA production. In the present study, feeding with a rumen-protected microencapsulated supplement (via linseed oil coated with hydrogenated palm oil) was able to prevent drastic modifications to the linseed oil. However, if the released linseed oil is present in the rumen for a longer time, it might increase the CLA content in milk. Conversely, the decrease of CLA in the LO group seemed to be due to reduced DMI (Table 6). Before performing the in vivo experiments, it was assumed that the levels of omega-6 fatty acids in the milk would be decreased, while the levels of omega-3 fatty acids would be increased in the milk due to the addition of omega-3 fatty acids. However, unexpectedly, C18:2n6c (linoleic acid) and C18:3n6 (γ-linolenic acid) showed no changes in the LO group, whereas the levels of C18:2n6t (linolelaidic acid) and C20:4n6 (arachidonic acid) were increased in the MO group compared to those in the LO group and those before treatment. According to Cattani et al. [28], the addition of omega-3 fatty acids in ruminants generally lowers omega-6 fatty acids in their products and promotes omega-3 fatty acids, thus affecting the omega-6:omega-3 ratios. The results of the present study suggest that omega-6 fatty acids showed no reduction in the LO group because of decreased DMI (Table 6 and Table 7). However, the increase of omega-6 fatty acids in the MO group might have been due to the transformation of linoleic acid in linseed oil to arachidonic acid in the mammary glands [29]. The reason for the increase of linolelaidic acid in the MO group is currently unclear. Among the omega-3 fatty acids, C18:3n3 (ALA) was increased in both the MO and LO groups compared to that in the control, although it showed a greater increase in the MO group than in the LO group (increased value of ALA, 2-fold higher: 13.98 vs. 5.15 mg/100 g milk), possibly due to a reduced DMI in the LO group. However, this result suggests that ALA can bypass the rumen more safely due to the ruminal protection effect of MO, which is superior to the protection effect of general linseed oil [2,30]. As a result, the MO supplement induced a greater yield of total omega-3 fatty acid than LO (8.87 vs. 17.77 mg/100 g milk), including a greater decrease in the ratio of omega-6 to omega-3 fatty acids in the MO group (7.32 vs. 3.48). Collectively, these results suggest that the dietary supplementation of 2% MO is beneficial for fatty acid (omega-3) elevation without any negative effects on milk yield in dairy cows.

## 5. Conclusions

Here, we found that rumen-bypass microencapsulated fatty acids from linseed oil clearly and safely improved the pass rate of omega-3 fatty acids from the rumen to the small intestine. Moreover, their protection effect at a dosage of up to 2% had no negative or regressive influence on ruminal fermentation or animal productivity among the dairy cows. Indeed, it is arguable that the use of only one cow in the in vitro experiment produced a difference between the in vitro and in vivo diet results. However, this difference is not considered to have affected our results. Collectively, our results suggest a direction for increasing the levels of omega-3 fatty acids in ruminant milk.

## Figures and Tables

**Figure 1 animals-10-01631-f001:**
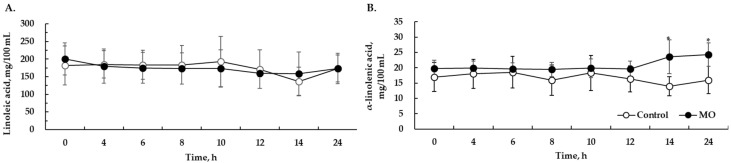
Blood concentrations of linoleic acid (**A**) and α-linolenic acids (**B**) from 0 to 24 h post feeding in cows. Control, no MO supplementation to basal diet. MO, 2% rumen-protected microencapsulated fatty acids from linseed oil supplementation to basal diet. Values are presented as the means ± standard errors (*n* = 4). * indicates a significant diet effect at *p* < 0.05.

**Table 1 animals-10-01631-t001:** Ingredients and chemical compositions of the diets in each treatment group.

Diet Ingredients for a Cow	Treatment Groups ^1^		
	Before	LO	MO
kg/d as dry matter (DM)			
Concentrate	4.496	4.496	4.496
Sudan grass silage	1.680	1.680	1.680
Total mixed ration (TMR)	21.864	21.864	21.864
Corn gluten powder	0.000	1.131	1.131
Linseed oil	0.000	0.209	0.000
Rumen-protected microencapsulated supplement	0.000	0.000	0.596
Chemical composition, %			
DM	65.90	70.46	71.75
Crude protein	10.84	10.85	10.85
Ether extract	2.89	3.59	4.80
Crude fiber	14.55	14.57	14.57
Crude ash	4.89	8.65	8.65
Neutral detergent fiber	28.30	28.30	28.30
Acid detergent fiber Long chain fatty acid composition, %	14.07	14.07	14.07
Palmitic acid methyl ester (C16:0)	17.100	17.147	17.954
Stearic acid methyl ester (C18:0)	3.305	3.336	3.588
Oleic acid methyl ester (C18:1n9c)	19.253	19.392	19.607
Linoleic acid methyl ester (C18:2n6c)	33.313	33.429	33.432
Linolenic acid methyl ester (C18:3n3)	5.264	5.635	5.630

^1^ Before, basal diet; LO, linseed oil group; MO, rumen-protected microencapsulated supplement group. Ingredients of the TMR are 1.5% molasses, 1% palm kernel meal, 2% soybean meal, 1.5% wheat bran, 1% rapeseed meal, 2% corn germ meal, 11.5% alfalfa hay, 2.5% beet pulp, 23% concentrate, 9% corn flake, 5% cotton seed, 5% kleingrass hay, 18.5% oaten hay, and 2.5% timothy hay. Concentrate ingredients are 9% canola seed meal, 6% coconut kernel meal, 20% corn, 0.2% dicalcium phosphate, 1.5% limestone, 0.2% MgO, 6% molasses, 14.2% palm kernel meal, 0.5% salt, 2% sesame seed meal, 0.4% sodium bicarbonate, 10% soybean meal, 2% tallow, 3% tapioca, 11% wheat middling, 9% canola seed meal, and 0.4% vitamin mineral premix (1,000,000 IU Vit. A, 100,000 IU Vit. D3, 25,000 Vit. E, 150 I, 150 Co, 2500 Cu, 6250 Fe, 16,000 Mn, 10,000 Zn, 150 mg Se).

**Table 2 animals-10-01631-t002:** Effects of the rumen-protected microencapsulated supplement (MO) on pH, gas production, and ammonia-N in the in vitro ruminal fluid.

Incubation Times, h	CON ^1^	MO Concentrations ^2^ (%)					SEM ^3^	*p*-Value		
		1	2	3	4	5		Diet	L ^4^	Q ^5^
pH										
0	6.75	6.79	6.77	6.77	6.74	6.73	0.012	0.051	0.094	0.076
12	5.77	5.80	5.79	5.78	5.76	5.75	0.011	0.053	0.064	0.109
24	5.65	5.58	5.60	5.60	5.57	5.59	0.019	0.080	0.135	0.327
48	5.54	5.54	5.56	5.54	5.55	5.53	0.009	0.156	0.522	0.349
Total gas, mL										
12	33.17	31.33	32.00	32.00	33.67	30.33	0.492	0.779	0.645	0.837
24	47.67	49.00	47.67	41.67	49.67	43.33	0.579	0.063	0.137	0.956
48	51.67 ^b^	58.33 ^a^	59.33 ^a^	57.67 ^a^	50.33 ^b^	50.00 ^b^	0.557	<0.001	<0.001	<0.001
CH_4,_ %										
12	11.15 ^c^	11.33 ^bc^	11.35 ^bc^	11.51 ^ab^	11.51 ^ab^	11.78 ^a^	0.053	0.001	<0.001	0.551
24	10.27 ^c^	12.55 ^bc^	12.55 ^bc^	12.64 ^abc^	12.88 ^ab^	13.12 ^a^	0.280	0.004	0.111	0.339
48	13.11	12.96	11.78	13.50	12.79	13.89	0.328	0.606	0.477	0.309
CH_4,_ mL ^6^										
12	8.77	8.70	8.80	8.92	9.11	8.94	0.079	0.769	0.239	0.945
24	11.48	11.86	11.68	11.01	11.50	11.65	0.111	0.148	0.611	0.433
48	12.74	13.46	12.36	13.93	12.26	13.26	0.330	0.711	0.934	0.915
Ammonia-N, mg/100 mL										
0	3.04 ^d^	3.17 ^cd^	3.27 ^abc^	3.39 ^ab^	3.44 ^a^	3.21 ^bcd^	0.026	<0.001	<0.001	<0.001
12	4.69 ^c^	4.71 ^c^	4.90 ^bc^	4.96 ^abc^	5.16 ^ab^	5.31 ^a^	0.047	<0.001	<0.001	0.407
24	10.74 ^abc^	10.72 ^abc^	10.81 ^ab^	11.09 ^a^	10.43 ^bc^	10.34 ^c^	0.055	<0.001	0.009	0.002
48	20.45	20.75	21.24	20.60	20.72	20.42	0.104	0.288	0.688	0.146

^1^ CON, no MO supplementation to basal diet. ^2^ MO, 1% to 5% MO supplementation to basal diet. ^3^ SEM, standard error of the mean. ^4^ L, linear. ^5^ Q, quadratic. ^6^ CH_4,_ mL, calculated as [gas production (total gas, mL) + initial blank volume (45.5 mL)] × CH_4_ (%). ^a–d^ Different superscripts represent differences among treatments (*p* < 0.05).

**Table 3 animals-10-01631-t003:** Effects of rumen-protected microencapsulated supplement (MO) on volatile fatty acid (VFA) production in the in vitro ruminal fluid.

Items	CON ^1^	MO Concentrations ^2^, %					SEM ^3^	*p*-Value		
		1	2	3	4	5		Diet	L ^4^	Q ^5^
12 h										
Total VFA, mM	53.80	51.64	52.41	53.93	55.45	51.63	0.550	0.610	0.521	0.180
Acetic acid, C2	26.47	25.48	25.75	26.52	27.29	25.42	0.213	0.595	0.517	0.178
Propionic acid, C3	8.25	7.94	8.04	8.28	8.48	7.86	0.127	0.583	0.704	0.185
Iso-butyric acid, C4	0.19	0.18	0.19	0.19	0.20	0.19	0.022	0.834	0.481	0.378
Butyric acid, C4	16.31	15.59	15.92	16.37	16.85	15.70	0.193	0.660	0.456	0.182
Iso-valeric acid, C5	1.28	1.21	1.24	1.28	1.30	1.22	0.060	0.518	0.463	0.188
Valeric acid, C5	1.30	1.24	1.27	1.29	1.33	1.24	0.057	0.518	0.652	0.252
A:P, C2/C3	3.21	3.21	3.20	3.20	3.22	3.23	0.012	0.672	0.121	0.118
24 h										
Total VFA, mM	60.60	60.93	59.14	55.91	58.09	57.14	0.661	0.191	0.104	0.138
Acetic acid, C2	28.79	29.04	28.12	26.68	27.66	27.29	0.244	0.27	0.117	0.141
Propionic acid, C3	8.62	8.73	8.40	7.96	8.26	8.15	0.137	0.226	0.091	0.092
Iso-butyric acid, C4	0.22	0.23	0.22	0.20	0.21	0.21	0.022	0.408	0.084	0.067
Butyric acid, C4	19.92	20.30	19.42	18.25	19.00	18.61	0.228	0.215	0.041	0.045
Iso-valeric acid, C5	1.59	1.13	1.54	1.47	1.53	1.49	0.269	0.461	0.301	0.374
Valeric acid, C5	1.47	1.50	1.44	1.36	1.42	1.39	0.062	0.298	0.070	0.084
A:P, C2/C3	3.34	3.33	3.35	3.35	3.35	3.35	0.011	1.000	0.287	0.284
48 h										
Total VFA, mM	63.22	64.75	66.27	65.38	65.91	65.78	0.335	0.231	0.238	0.265
Acetic acid, C2	29.08	29.58	30.35	29.95	30.21	30.02	0.111	0.244	0.284	0.155
Propionic acid, C3	8.53	8.77	9.05	8.89	8.99	8.92	0.077	0.205	0.359	0.22
Iso-butyric acid, C4	0.26 ^b^	0.26 ^b^	0.27 ^a^	0.27 ^a^	0.27 ^a^	0.27 ^a^	0.012	0.036	0.028	0.098
Butyric acid, C4	21.71	22.41	22.77	22.51	22.60	22.75	0.119	0.232	0.440	0.749
Iso-valeric acid, C5	2.00 ^b^	2.05 ^ab^	2.10 ^ab^	2.07 ^ab^	2.10 ^ab^	2.11 ^b^	0.034	0.003	0.020	0.075
Valeric acid, C5	1.65	1.68	1.73	1.70	1.74	1.71	0.033	0.228	0.151	0.063
A:P, C2/C3	3.41 ^a^	3.37 ^ab^	3.35 ^b^	3.37 ^ab^	3.36 ^b^	3.37 ^ab^	0.014	0.004	0.922	0.759

^1^ CON, no MO supplementation to basal diet. ^2^ MO, 1% to 5% MO supplementation to basal diet. ^3^ SEM, standard error of the mean. ^4^ L, linear. ^5^ Q, quadratic. ^a–d^ Different superscripts represent differences among treatments (*p* < 0.05).

**Table 4 animals-10-01631-t004:** Digestibility of 0.7% linseed oil (LO) and 2% rumen-protected microencapsulated supplement (MO) in the in vitro ruminal fluid.

Digestibility, %	CON ^1^	0.7% LO ^2^	2% MO ^3^	SEM ^4^	*p*-Value
12 h					
Dry matter, %	33.34	31.42	31.82	1.609	0.905
NDF ^5^, %	58.38	56.77	58.56	1.448	0.753
ADF ^6^, %	55.17	58.83	56.93	1.758	0.888
Crude protein, %	6.87	6.85	8.62	0.970	0.790
Ether extract, %	29.89	22.83	15.93	4.111	0.483
24 h					
Dry matter, %	39.49	36.97	39.43	1.762	0.843
NDF, %	69.01	66.11	68.32	1.226	0.990
ADF, %	53.05	53.37	52.41	2.455	0.665
Crude protein, %	16.51	18.35	17.21	0.929	0.806
Ether extract, %	33.05	31.04	24.86	2.560	0.499
48 h					
Dry matter, %	48.04	45.13	46.60	1.289	0.716
NDF, %	72.39	70.49	72.71	0.783	0.986
ADF, %	56.52	56.94	57.31	1.687	0.521
Crude protein, %	19.40	20.66	25.68	2.858	0.742
Ether extract, %	38.15	34.91	32.78	3.384	0.875

^1^ CON, no LO or MO supplementation to basal diet. ^2^ LO, 0.7% linseed oil supplementation to basal diet. ^3^ MO, 2% rumen-protected microencapsulated fatty acids from linseed oil supplementation to basal diet. ^4^ SEM, standard error of the mean. ^5^ NDF, neutral detergent fiber. ^6^ ADF, acid detergent fiber.

**Table 5 animals-10-01631-t005:** Changes in fatty acid compositions after co-culturing 0.7% linseed oil (LO) and 2% rumen-protected microencapsulated supplement (MO) with ruminal fluid.

Fatty Acids	CON ^1^	0.7% LO ^2^	2% MO ^3^	SEM ^4^	*p*-Value
0 h					
Stearic acid, %	19.795 ^b^	19.079 ^b^	22.961 ^a^	0.625	0.018
Linoleic acid, %	14.283 ^a^	12.873 ^ab^	10.199 ^b^	0.761	0.023
α-linolenic acid, %	1.575 ^b^	8.567 ^a^	4.835 ^ab^	1.119	0.005
Conjugated linoleic acid (CLA), %	-	-	-		
Linoleic acid, mg/g	0.175	0.157	0.143	0.011	0.540
α-linolenic acid, mg/g	0.035 ^b^	0.113 ^a^	0.081 ^ab^	0.013	0.012
CLA, mg/g	-	-	-		
12 h					
Stearic acid, %	33.971	33.650	32.612	0.240	0.056
Linoleic acid, %	4.383	5.637	4.550	0.267	0.120
α-linolenic acid, %	0.371 ^b^	1.468 ^ab^	3.794 ^a^	0.506	<0.001
CLA, %	1.141 ^a^	0.992 ^a^	0.789 ^b^	0.054	<0.001
Linoleic acid, mg/g	0.061	0.082	0.091	0.008	0.274
α-linolenic acid, mg/g	0.022 ^b^	0.034 ^b^	0.082 ^a^	0.010	0.003
CLA, mg/g	0.018	0.017	0.019	0.001	0.917
24 h					
Stearic acid, %	38.790 ^ab^	40.473 ^a^	38.281 ^b^	0.368	0.040
Linoleic acid, %	1.758	2.215	2.146	0.103	0.133
α-linolenic acid, %	0.000 ^c^	0.531 ^b^	1.307 ^a^	0.192	<0.001
CLA^5^, %	0.000 ^b^	0.621 ^a^	0.666 ^a^	0.112	<0.001
Linoleic acid, mg/g	0.036	0.040	0.049	0.004	0.434
α-linolenic acid, mg/g	0.018 ^b^	0.023 ^ab^	0.040 ^a^	0.004	0.032
CLA, mg/g	0.000 ^b^	0.011 ^a^	0.015 ^a^	0.003	0.014
48 h					
Stearic acid, %	42.133 ^ab^	43.182 ^a^	40.432 ^b^	0.408	0.012
Linoleic acid, %	1.227 ^c^	1.473 ^b^	1.755 ^a^	0.077	<0.001
α-linolenic acid, %	0.000 ^c^	0.452 ^b^	1.250 ^a^	0.214	0.006
CLA, %	-	-	-		
Linoleic acid, mg/g	0.031	0.033	0.046	0.003	0.058
α-linolenic acid, mg/g	0.018 ^b^	0.022 ^b^	0.041 ^a^	0.004	<0.001
CLA, mg/g	-	-	-		

^1^ CON, no LO or MO supplementation to basal diet. ^2^ LO, 0.7% linseed oil supplementation to basal diet. ^3^ MO, 2% rumen-protected microencapsulated fatty acids from linseed oil supplementation to basal diet. ^4^ SEM, standard error of the mean. ^a–d^ Different superscripts represent differences among treatments (*p* < 0.05).

**Table 6 animals-10-01631-t006:** Effects of 0.7% linseed oil (LO) and 2% rumen-protected microencapsulated supplement (MO) on the dry matter intake (DMI) and milk compositions in dairy cows.

Items	LO ^1^		MO ^2^		SEM ^3^	*p*-Value		
	Before ^4^	After ^5^	Before	After		Fat Source	Period	F × P
DMI, kg/d/head	28.04	27.64	28.04	28.04	0.021	-	-	-
Milk yield, kg/d	31.79	29.74	33.30	30.54	1.548	0.459	0.124	0.819
Milk yield: DMI ratio	1.13	1.08	1.19	1.09	0.055	0.548	0.162	0.714
Fat, %	4.84	5.00	4.89	5.01	0.154	0.871	0.375	0.900
Fat, kg/d	1.53	1.48	1.62	1.51	0.080	0.450	0.313	0.745
4% FCM ^6^, kg/d	35.73	34.08	37.65	34.92	1.745	0.433	0.214	0.760
4% FCM: DMI ratio	1.27	1.23	1.34	1.25	0.062	0.521	0.271	0.656
ECM ^7^, kg/d	38.42	36.65	40.07	37.07	0.895	0.567	0.190	0.735
ECM: DMI ratio	1.37	1.33	1.43	1.32	0.032	0.671	0.245	0.629
Protein, %	3.59	3.66	3.47	3.47	0.074	0.034	0.652	0.638
Protein, kg/d	1.13	1.08	1.14	1.04	0.041	0.653	0.072	0.600
Lactose, %	4.72	4.77	4.72	4.75	0.039	0.862	0.492	0.765
Solid non-fat, kg/d	1.13	1.08	1.14	1.04	0.129	0.710	0.088	0.724
Milk urea nitrogen, mg/dL	15.51	13.61	15.23	10.80	0.385	<0.001	<0.001	<0.002

^1^ LO, 0.7% linseed oil supplementation to basal diet. ^2^ MO, 2% rumen-protected microencapsulated fatty acids from linseed oil supplementation to basal diet. ^3^ SEM, standard error of the mean. ^4^ Before, average data of d 0 and d 30 (period on feeding basal diet). ^5^ After, average data of d 30 and d 60 (period on feeding basal diet and oil supplements). ^6^ FCM, fat corrected milk; [0.40 × milk yield (kg/d)] + [15 × milk fat yield (kg/d)] (Gaines and Davidson, 1923). ^7^ ECM, energy corrected milk; [0.327 × milk yield (kg/d)] + [12.95 × fat yield (kg/d)] + [7.2 × protein yield (kg/d)]. Data were the covariates adjusted based on the initial data.

**Table 7 animals-10-01631-t007:** Effects of 0.7% linseed oil (LO) and 2% rumen-protected microencapsulated supplement (MO) on the complete blood count analysis in dairy cows.

Items ^6^	LO ^1^		MO ^2^		SEM ^3^	*p*-Value		
	Before ^4^	After ^5^	Before	After		Fat Source	Period	F × P
Hematological parameters								
LYM (2.5–7.5 K/uL)	6.95	9.32	6.88	8.67	0.587	0.764	0.083	0.806
MON (0–0.84 K/uL)	0.91	0.68	0.79	0.53	0.084	0.433	0.152	0.927
GRA (0.6–6.7 K/uL)	3.23	2.04	2.42	1.47	0.208	0.073	0.007	0.757
WBC (4–12 K/uL)	11.09	12.04	10.09	10.67	0.737	0.440	0.617	0.903
RBC (5–10 M/uL)	6.48	5.48	6.33	5.30	0.180	0.637	0.004	0.961
HGB (8–15 g/dL)	11.64	12.32	11.63	11.78	0.285	0.641	0.483	0.653
HCT (24–46%)	31.33	26.82	30.60	25.77	0.762	0.522	0.002	0.910
MCV (40–60 fL)	48.50	48.90	48.70	49.50	0.720	0.790	0.690	0.894
MCH (11–17 pg)	17.91	22.49	18.53	22.55	0.427	0.518	<0.001	0.594
MCHC (30–36 g/dL)	36.95	45.99	38.01	45.76	0.745	0.531	<0.001	0.332
Platelet (100–800 K/uL)	277.30	198.90	266.40	221.60	17.705	0.868	0.089	0.637
Metabolic parameters								
BUN, mg/dL	19.5	17.9	19.1	16.1	0.39	0.123	0.002	0.291
GLU, mg/dL	70.40	69.9	70.6	67.1	1.18	0.582	0.399	0.526
TCHO, mg/dL	320.2	345.9	314.0	365.2	9.68	0.718	0.052	0.518
NEFA, mg/dL	185.2	140.0	243.4	267.6	13.16	<0.001	0.632	0.120

^1^ LO, 0.7% linseed oil supplementation to basal diet. ^2^ MO, 2% rumen-protected microencapsulated fatty acids from linseed oil supplementation to basal diet. ^3^ SEM, standard error of the mean. ^4^ Before, average data of d 0 and d 30 (period on feeding basal diet). ^5^ After, average data of d 30 and d 60 (period on feeding basal diet and oil supplements). ^6^ Items = LYM, lymphocyte; MON, monocyte; GRA, granulocyte; WBC, white blood cell; RBC, red blood cell; HGB, hemoglobin; HCT, hematocrit; MCV, mean corpuscular volume; MCH, mean corpuscular hemoglobin; MCHC, mean corpuscular hemoglobin concentration; BUN, blood urine nitrogen; GLU, glucose; TCHO, total cholesterol; NEFA, non-esterified fatty acid.

**Table 8 animals-10-01631-t008:** Effects of 0.7% linseed oil (LO) and 2% rumen-protected microencapsulated supplement (MO) on milk FA compositions in dairy cows.

Items ^6^	LO ^1^		MO ^2^		SEM ^3^	*p*-Value		
	Before ^4^	After ^5^	Before	After		Fat Source	Period	F × P
C16:0 (Palmitic), %	32.77	31.84	32.20	32.85	0.280	0.700	0.796	0.164
C18:0 (Stearic), %	13.24	15.56	13.87	16.84	0.280	0.040	<0.001	0.476
C18:1 (TVA), %	0.71	0.75	0.70	0.88	0.029	0.143	0.006	0.092
CLA, mg/100 g	11.01	10.00	10.71	11.72	0.427	0.243	0.999	0.099
C18:2n6t, mg/100 g	6.43	7.92	6.61	10.58	0.304	0.002	<0.001	0.008
C18:2n6c, mg/100 g	50.02	50.83	48.76	40.88	1.087	0.005	0.073	0.028
C18:3n6, mg/100 g	1.08	1.13	1.13	1.15	0.011	0.092	0.108	0.381
C20:3n6, mg/100 g	6.06	5.88	6.31	5.31	0.111	0.439	0.005	0.048
C20:4n6, mg/100 g	0.54	0.58	0.59	0.65	0.009	<0.001	0.004	0.285
C18:3n3, mg/100 g	5.15	5.83	4.99	13.98	0.571	<0.001	<0.001	<0.001
C20:3n3, mg/100 g	0.94	0.91	1.11	1.03	0.040	0.055	0.487	0.780
C20:5n3, mg/100 g	0.97	0.97	1.09	0.92	0.017	0.255	0.008	0.005
C22:6n3, mg/100 g	1.82	1.81	2.06	1.84	0.040	0.082	0.145	0.184
SFA, %	77.09	77.55	75.91	76.22	0.311	0.045	0.533	0.896
MUFA, %	18.42	17.89	19.75	19.30	0.298	0.022	0.402	0.947
PUFA, %	4.49	4.56	4.34	4.49	0.062	0.367	0.392	0.740
SFA/UFA	3.42	3.52	3.19	3.27	0.063	0.064	0.458	0.947
ω6, mg/100 g	64.14	66.33	63.40	58.58	1.187	0.064	0.563	0.125
ω3, mg/100 g	8.87	9.51	9.24	17.77	0.603	<0.001	<0.001	<0.001
ω6 / ω3	7.32	6.99	6.89	3.48	0.204	<0.001	<0.001	<0.001

^1^ LO, 0.7% linseed oil supplementation to basal diet. ^2^ MO, 2% rumen-protected microencapsulated fatty acids from linseed oil supplementation to basal diet. ^3^ SEM, standard error of the mean. ^4^ Before, average data of d 0 and d 30 (period on feeding basal diet). ^5^ After, average data of d 30 and d 60 (period on feeding basal diet and oil supplements). ^6^ Items = CLA, conjugated linoleic acid; SFA, saturated fatty acid; MUFA, mono-unsaturated fatty acid; PUFA, poly-unsaturated fatty acid. Data were covariates adjusted from initial data.

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
