# Peer review of "In Vitro and In Vivo Studies of Rumen-Protected Microencapsulated Supplement Comprising Linseed Oil, Vitamin E, Rosemary Extract, and Hydrogenated Palm Oil on Rumen Fermentation, Physiological Profile, Milk Yield, and Milk Composition in Dairy Cows"

_animals, 2020, doi:10.3390/ani10091631_

Round 1

Reviewer 1 Report

I read the revised work and noticed that in my opinion many critical issues have not been resolved, for example: the abstract should be corrected because at lines 32-33 the sentence contains incorrect indications. Digestibility of CP and EE does not consider the microbial contamination, therefore it is not correct to present it in this way. It is not possible to present the statistical analysis of the DMI when individual ingestion has not been measured (they only had group ingestion). There is confusion about TMR and basal diet, see abstract and lines 161-163. They must make it clear what the cows ate. The percentage of fat in the milk is very high for Holstein, they should check the data or the analysis method. Some tables are not clear.

Reviewer 2 Report

The manuscript by Kim et al is interesting. The authors used both in vitro and in vivo ruminal models to demonstrate effects of omega-3 fatty acids coated with hydrogenated palm oil (MO) on 1) pH, gas production, ammonia-N, VFAs in the rumen fluid; and 2) FA compositions in both rumen fluid and milk and milk compositions. Significant findings were reported, however, several major concerns need to be addressed. Firstly, what is the rationale to use 2% MO from in vitrostudy as an optimal dosage of MO supplementation for the follow-up in vivo study? The optimal dosage of MO for in vivostudies should be determined by pre- in vivo studies instead of using results from in vitro as a reference.  Secondly, it is unclear how many cows per group were used in the period 2 treatment? Why most in vivo data were based on four cows per group? How did you select these cows? Are they randomly selected or are you picking up cows in a biased manner? Thirdly, some baseline data were missing. For example, in Table 2, why 0 h data were not reported for total gas and CH4? Fourthly, a technical question: it is very difficult to distinguish isomers (e.g., iso-butyric acid vs butyric acid, linoleic acid vs α-linoleic acid, etc.). You will need to give detailed information about how did you annotate these isomers in the measurements. Fifthly, clinical observation data of cows during the experimental period are missing. Are all cows clinically healthy? Any diseases were recorded? In addition, the tittle needs to be revised. It is not clear about ‘what does improved milk fatty acids mean?’

Reviewer 3 Report

Comments.

Dietary supplementation of omega 3 coated with hydrogenated palm oil improves milk fatty acids in Holstein dairy cows

Detailed comments

Title

The statement “omega 3 coated with hydrogenated palm oil” it is not certainly correct. You only coated linseed oil (35%), vitamin E (0.5%) and rosemary extract (0.3%). This is not pure omega 3 fatty acids. This should change.

“In vitro” an “in vivo” models specify a big part of the manuscript. I suggest to include them in the title. You measured other variables apart from milk fatty acids, please consider to include the most relevant variables to better describe your investigation.

Simple summary,

Though milk fatty acid composition has important implications for human health, your paper is not dealing with it directly. Focus this section to your most important findings.

Introduction

Line 72-76. The hypothesis and the objective of the study are not very well connected. Please rephrase the final part of this section to match both ideas.

Materials and methods

Line 83 Only one animal was sampled to collect ruminal liquid? I this is true, explain the possible implications of this limitation. The offered diet to this(these) animal(s) was different from the diets evaluated in the in vivo and in vitro models. Elaborate a little bit in the conclusions section the possible implications about having different diets.

Alfalfa and concentrate VS Concentrate, sudan grass, corn gluten powder etc.

In the study you used 0.7% linseed oil or 2% MO (as DM basis). How did you decide to use this proportions? It is not explained. Both are equivalents?

64.2% of MO is hydrogenated palm oil versus 35% of linseed oil. It is possible that the effect of palm oil would be grater that linseed oil on the MO treatment?

Ether extract percentage for each treatment were: 2.89, 3.59 and 4.80 %. Explain how this would affect fatty acid profile and ruminal kinetics.

Please mention the volume of milk collected for analysis.

For fatty acid analysis you did not mention the standards utilized. Please include them. Which CLA isomers did you measured?

Statistical analysis

Indicate the statistical models used: Yij = μ + ……..

Results

It is common to use the literal “a” for the highest value and the rest of the literals for the other values. Here, I do not see this pattern. Please modify through all the manuscript.

Title of table 2. Specify that this results belong to the in vitro model.

Discussion

There is a growing concern about the use of palm oil, because the cultivars have impacts on land degradation and losses of plant and animal diversity, among other effects. Should be sustainable the recommendation of include two percent of palm oil to diets of dairy animals to increase healthy fatty acid profile?

Reviewer 4 Report

Dear authors,

I had the pleasure to read and review your article. Your research was very complete; however, I did detect some things that need to be explained or could be improved.

- In the first paragraph of the introduction. It is not clear if the benefits are for humans, I think it would be convenient to be more specific.

- In ruminal fermentation studies in vitro, it is highly recommended to use ruminal fluid from at least two animals of the same species and carry out three fermentations (runs).

- It is necessary to mention the methodologies for in vitro assay. You mention that you used the McDougall, but how did you measure gas production (water displacement, manometer, etc.) or the reference?

- It is not clear why in the in vitro assay, the pH, gas production, methane and N-ammoniacal were analyzed with five concentrations of MO and the control diet, but for digestibility you compared linseed oil with MO.

- In the “analysis” section (lines 131 to 145), references to determine NDF, ADF, CP, EE, or digestibility are not mentioned. Since they are widely used methodologies, I consider that it is not necessary to give details, with the reference is sufficient. Nor do I consider it necessary to include the formulas.

- How many cows did they use to verify the absorption of omega 3?

- I think there is confusion regarding TMR. A TMR contains all the food and all the nutrients. You report an additional silage and concentrate to the TMR.

- It is essential that the ingredients of the cows' diet are mentioned.

- Table 1 is confusing. Ingredients are not mentioned (mentioned above). Nor is it mentioned whether these amounts are per cow, per day and whether they are on a dry basis or as feed. Another confusing aspect is fatty acids composition, are they in percentages?  As they are presented, it seems that they are percentages of the composition in total, which is not possible because it exceeds 100%. Lastly, I do not understand why the composition of the diet of period 1 reports fatty acid content similar to the diets with linseed oil and MO, since it does not report a source of lipids.

- Table 2 needs to be improved.

- It is unnecessary to report the time 0h in table 4.

- In Table 6, I do not understand why there are statistical differences in DMI if there is only a decrease of 400g in the linseed oil group and it is the same in MO; however, there are no differences in milk production, even when in one of the treatments there is a difference of up to 3 kg between periods. I consider it necessary to review your data well.

- Definitely, the conclusions should be changed taking into account exclusively the results obtained in the study. The first four lines are out of place.

Regards,

Round 2

Reviewer 1 Report

The statistical comparison on the ingested dry matter could do only if the individual ingestions are measured. If you have group ingestions it would be better to report the estimate of individual ingestion without statistical comparison.

Minor revision:

In table 3 the superscript letters for Iso-butyric acid, C4 and Iso-valeric acid, C5 are missing.

For A: P, C2 / C3 it is surprising to find P = 1.000 with not all the same data, it should be verified.

In table 5 the superscript letters are missing for α-linolenic acid, and reduce to 3 the decimal places of the SEM.

Reviewer 4 Report

Dear authors,

Again, I have to say that very complete work, so it is worth doing everything possible to publish it. However, I still have several questions:

- There is confusion between TMR, concentrate and basal diet. The basal diet, as its name indicates, is the base diet that animals consumed. A TMR is a total mixed ration, which means that it includes all the ingredients, and all the nutrients that animals require. And a concentrate is a kind of feedstuff that includes certain ingredients and certain nutrients that complement the rest of the diet. If you used a TMR, other foods such as silage and concentrate should no longer be included. Please clarify this aspect.

- In lines 92 to 109, in vitro fermentation is described. Mention the reference and how you measured gas production. It is not clear whether in vitro fermentation was for the basal diet or only for the TMR.

- In line 123 mention the reference for measurement of VFA by gas chromatography

- In lines 125-127 it is necessary to check the reference. To my knowledge, AOAC does not describe the methodology for digestibility of NDF, ADF, CP, EE, etc. AOAC mentions how CP, EE, etc. is determined. Also, mention the reference for NDF and ADF.

 - In lines 129-144, eliminate the formulas to obtain the content of CP, EE, etc.

- In Table 1, the composition exceeds 100%. Review how the fatty acid values are given.

- Neither the text nor Table 1 mentions the ingredients of the TMR and the concentrate. It is essential to include them.

- In Table 2, the CH4 mL do not correspond to %. Mention if the mL of total gas and CH4 are per g of DM or how they are reported.

Regards,

Author Response

This manuscript is a resubmission of an earlier submission. The following is a list of the peer review reports and author responses from that submission.

Round 1

Reviewer 1 Report

I can't fully evaluate the manuscript for several reasons:

- the work refers to 5 tables (S1-S5) which are not reported and therefore it is not possible to see the data mentioned in the text;

- in the materials and methods a part of the in vitro work is not presented (digestibility with 0.7 of LO or 2% of MO, tab 3 and 4) which is instead exposed in the results and commented in the discussion for which this part of the work is not acceptable;

- in materials and methods reference is made to 8 treatments and in tables and in the results there are only 6;

- in materials and methods, statistical analysis, tables are mentioned that do not appear in the text (tables S4, S5 and table 10). In addition, the description given is not clear: different periods result in the "oil" comparison of the tables, but the period effect is not considered. Furthermore, there is an interaction in some tables (Coat x Oil) not described. In some tables Linear and Quadratic P-values ​​are reported, not even mentioned in the materials and methods paragraph, 3 P-values ​​(Diet, Linear, Quadratic) are found which are not explained;

- in lines 140-142: first it is said that the cows ingested the TMR and in the following sentence they ate the total mixed ration (78%), ensiled (6) and concentrated (16%). This is not clear!

- Tables 5, 6 and 7 indicate a P-value for the "Oil" treatment, which means that it is compared with the previous period without Oil, but being different periods they are not comparable in this way;

- table 5 shows the DMI with its statistical analysis, but the materials and methods do not explain how the individual ingestion of the cows was determined;

- in the materials and methods it is not clear how the milk was sampled for subsequent analyzes, it is essential to provide this information in detail.

Furthermore, in the results and in the summary, differences are declared that are not supported by statistical analysis (p> 0.05):

for example on lines 32-33 it is written that the MO treatment had negative effects only at concentrations of 3, 4 and 5% with less total gas produced and greater concentration of ammonia, but between 2% and 3% there are no statistically significant differences for these 2 parameters. Again, in lines 192-193 it is said that the methane proportions (%) were higher in the treatments from 1 to 5% compared to the Control, but this is true only for treatment 3, 4 and 5% at 12 h, only for treatment 4 and 5% at 24 h and never for 48 h.

The meaning of the superscript letters is not shown or explained in several tables.

In light of the foregoing, it is evident that before being able to proceed with an accurate review of the work, it is necessary to rewrite the work without such flaws.

Author Response

Dear sir. Reviewer,

Please find enclosed the revised version of our manuscript entitled “Dietary supplementation of omega 3 coated with hydrogenated palm oil improves milk fatty acids in Holstein dairy cows”

We appreciate your all valuable comments and trust that we have made all of the changes necessary again.

Sincerely,

Hong-Gu LEE, the corresponding author (hglee66@konkuk.ac.kr)

Reviewer 2 Report

I cant get my author comments to load. I'll send them separately.

Author Response

(The authors gave the same response as above.)

Round 2

Reviewer 1 Report

The manuscript is not acceptable in this form as it has not been improved since the first revision.

- Tables S1, S2, S3, S4 and S5 mentioned in the text are not yet reported in the manuscript. If adding 5 tables were excessive, the authors must find ways to report the data in a lower number of added tables, but I cannot say more since I do not see these tables (moreover the authors have declared to have loaded the supplementary tables, but I have not found them).

- The abstract should be corrected because lines 32-33 report that the MO treatment had negative effects only at concentrations of 3, 4 and 5% with less total gas produced and greater concentration of ammonia, but between 2% and 3% there are no statistically significant differences for these 2 parameters.

- It is necessary to describe the sampling method for determining the total gas and methane production.

- In Table 3 you talk about digestibility of NDF, ADF, CP and EE, but in the materials and methods there is a description of the digestibility only of NDF and no other methods are described. I believe that in table 3 there are only the data of the analyses of ADF, CP and EE of the incubation residues, but this is not digestibility. The added materials and methods (lines 107-110) are not clear and it should be described how the digestibility of NDF, ADF, CP and EE was determined. With the Ankom (Daisy?) Method, only the digestibility of the NDF can be determined. I recommend to report only the digestibility data of the NDF and to indicate if it is the IVTD (in vitro true digestibility) of Ankom made with Daisy, as I believe.

The whole new part reported in lines 126-139 is the method for determining the NDF or ADF content of a sample, but not their digestibility.

- It is not acceptable that superscripts have been removed from the tables. In this way it is not possible to understand the results. The superscripts letters must be put and commented correctly, for example in lines 192-193 it is said that the methane proportions (%) were higher in the treatments from 1 to 5% compared to the Control, but this is true only for treatment 3, 4 and 5% at 12 h, only for treatment 4 and 5% at 24 h and never for 48 h.

- In line 30 it is said that cows eat only TMR, then in lines 159-160 it is reported that they eat the TMR (78%) more silage (6%) and concentrated (16%). I believe it is correct in the abstract as the TMR should be the TOTAL mixed ration. However, you must make it clear what the cows ate.

- Please describe the system for measuring individual ingestion of cows (table 5 shows the DMI with its statistical analysis, but the materials and methods do not explain how the individual ingestion of the cows was determined).

- Please describe the system used to measure and sample individual milk production.

- In table 5 the data of fat percentage reported are very high for Holstein cows, need an explanation.

In light of the above comments, I still have to say that in my opinion the work cannot be published without a major revision.

Author Response

(The authors gave the same response as above.)

Reviewer 2 Report

Statistics are still not clear. you made it clear in your responses that when you did the in vitro fermentations that you took subsamples from each flask at different times of incubation. This requires a statistical model of analysis that includes a main plot of treatments and a subplot of time of sampling that each have different degrees of freedom. If this is not done correctly it could lead to wrong conclusions about the effects of treatments.

you explanation to my question about subsampling was mis-interpreted I think as you answered quesyions about how fermentation was stiopped rather than address the statistics.

you would be best to write out the model used in the statistical model showing all sources of variation used.

Author Response

(The authors gave the same response as above.)
